# Brain atlas for glycoprotein hormone receptors at single-transcript level

**Vitaly Ryu[1,2], Anisa Gumerova[1,2], Funda Korkmaz[1,2], Seong Su Kang[3], Pavel Katsel[4], Sari Miyashita[1,2], Hasni Kannangara[1,2], Liam Cullen[1,2], Pokman Chan[5], TanChun Kuo[1,2], Ashley Padilla[1,2], Farhath Sultana[1,2], Soleil A Wizman[1], Natan Kramskiy[1], Samir Zaidi[6], Se-Min Kim[1,2], Maria I New[7], Clifford J Rosen[8], Ki A Goosens[1,4], Tal Frolinger[1,2], Vahram Haroutunian[4], Keqiang Ye[9], Daria Lizneva[1,2], Terry F Davies[1,2], Tony Yuen[1,2]\*, Mone Zaidi[1,2]\***

[1]Center for Translational Medicine and Pharmacology, Icahn School of Medicine at Mount Sinai, New York, United States; [2]Department of Medicine and of Pharmacological Sciences, Icahn School of Medicine at Mount Sinai, New York, United States; [3]Department of Pathology, Emory University School of Medicine, Atlanta, United States; [4]Department of Psychiatry, Icahn School of Medicine at Mount Sinai, New York, United States; [5]Alamak Biosciences, Beverly, United States; [6]Memorial Sloan Kettering Cancer Center, New York, United States; [7]Department of Pediatrics, Icahn School of Medicine at Mount Sinai, New York, United States; [8]Maine Medical Center Research Institute, Scarborough, United States; [9]Faculty of Life and Health Sciences, and Brain Cognition and Brain Disease Institute, Shenzhen Institute of Advanced technology, Chinese Academy of Sciences, Shenzhen, China

**\*For correspondence:**
Tony.yuen@mountsinai.org (TY);
mone.zaidi@mountsinai.org (MZ)

**Abstract** There is increasing evidence that anterior pituitary hormones, traditionally thought to have unitary functions in regulating single endocrine targets, act on multiple somatic tissues, such as bone, fat, and liver. There is also emerging evidence for anterior pituitary hormone action on brain receptors in mediating central neural and peripheral somatic functions. Here, we have created the most comprehensive neuroanatomical atlas on the expression of TSHR, LHCGR, and FSHR. We have used RNAscope, a technology that allows the detection of mRNA at single-transcript level, together with protein level validation, to document *Tshr* expression in 173 and *Fshr* expression in 353 brain regions, nuclei and subnuclei identified using the *Atlas for the Mouse Brain in Stereotaxic Coordinates*. We also identified *Lhcgr* transcripts in 401 brain regions, nuclei and subnuclei. Complementarily, we used ViewRNA, another single-transcript detection technology, to establish the expression of *FSHR* in human brain samples, where transcripts were co-localized in *MALAT1*-positive neurons. In addition, we show high expression for all three receptors in the ventricular region—with yet unknown functions. Intriguingly, *Tshr* and *Fshr* expression in the ependymal layer of the third ventricle was similar to that of the thyroid follicular cells and testicular Sertoli cells, respectively. In contrast, *Fshr* was localized to NeuN-positive neurons in the granular layer of the dentate gyrus in murine and human brain—both are Alzheimer's disease-vulnerable regions. Our atlas thus provides a vital resource for scientists to explore the link between the stimulation or inactivation of brain glycoprotein hormone receptors on somatic function. New actionable pathways for human disease may be unmasked through further studies.

## Editor's evaluation

This article is an excellent resource as an atlas of hypophyseal hormone localization in the brain. It is an invaluable resource to researchers in the field and provides important new information.

## Introduction

There is increasing evidence that pituitary hormones traditionally thought of as 'pure' regulators of single physiological processes affect multiple bodily systems, either directly or via actions on brain receptors (*Zaidi et al., 2018*; *Abe et al., 2003*). We established, for the first time, a direct action of thyroid-stimulating hormone (TSH) on bone and found that TSH receptor (TSHR) haploinsufficiency causes profound bone loss in mice (*Abe et al., 2003*). We also found that follicle-stimulating hormone (FSH), *hitherto* thought to solely regulate gonadal function, displayed direct effects on the skeleton to cause bone loss (*Sun et al., 2006*), and on fat cells, to cause adipogenesis and body fat accumulation (*Liu et al., 2017*). Likewise, we showed that hormones from the posterior pituitary, namely, oxytocin and vasopressin, displayed direct, but opposing, skeletal actions—effects that may relate to the pathogenesis of bone loss in pregnancy and lactation, and in chronic hyponatremia, respectively (*Sun et al., 2019*; *Sun et al., 2016*; *Tamma et al., 2009*; *Tamma et al., 2013*). To add to this complexity, and in addition to the poorly recognized ubiquity of pituitary hormone receptors, the ligands themselves, or their variants, are expressed widely. We find the expression of a TSHβ variant (TSHβv) in bone marrow macrophages, while oxytocin is expressed by both osteoblasts and osteoclasts (*Colaianni et al., 2011*; *Colaianni et al., 2012*; *Baliram et al., 2013*; *Baliram et al., 2016*). These studies have together shifted the paradigm from established unitary functions of pituitary hormones to an evolving array of yet unrecognized roles of physiological and pathophysiological importance.

There is a compelling body of literature to support the expression of oxytocin receptors in various brain regions, and their function in regulating peripheral actions, such as social behavior and satiety (*Sun et al., 2019*; *Bale et al., 2001*). However, there is relatively scant information on the expression and, importantly, function of the anterior pituitary glycoprotein hormone family of receptors, namely, FSHR, TSHR, and luteinizing hormone/human chorionic gonadotropin receptor (LHCGR). Discrete sites of the rat, mouse, and human brain express receptors for these hormones, with several studies pointing to their relationship to neural functions, such as cognition, learning, neuronal plasticity, and sensory perception, as well as to neuropsychiatric disorders, including affective disorders and neurodegeneration (*Crisanti et al., 2001*; *Emanuele et al., 1985*; *Lei et al., 1993*; *Luan et al., 2020*; *Bi et al., 2020*; *Blair et al., 2019*; *Apaja et al., 2004*; *Naicker and Naidoo, 2018*; *Table 1*). In the light of such discoveries, the link between the stimulation of these receptors in the brain and the regulation of peripheral physiological processes needs further investigation.

Here, we use RNAscope—a cutting-edge technology that detects single RNA transcripts—to create the most comprehensive atlas of glycoprotein hormone receptors in mouse brain. This compendium of glycoprotein hormone receptors in concrete brain regions and subregions at a single-transcript level should allow investigators to study both peripheral and central effects of the activation of individual receptors in health and disease. Our identification of brain nuclei with the highest density for each receptor should also create a new way forward in understanding the functional engagement of receptor-bearing nuclei within a large-scale functional network.

## Results

Very little is known about the function(s) of anterior pituitary hormone receptors in the brain, except for isolated studies showing a relationship with cognition and affect (*Table 1*). We therefore used RNAscope to map the expression of *Tshr*, *Lhcgr,* and *Fshr* in the mouse brain; immunofluorescence and qPCR to provide confirmatory evidence for *Tshr* and *Fshr* expression; and ViewRNA and qPCR to examine for *FSHR* expression in AD-vulnerable regions of the human brain. RNAscope, which allows the detection of single transcripts, uses ~20 pairs of transcript-specific double Z-probes to hybridize 10-μm-thick whole-brain sections. Preamplifiers first hybridize to the ~28-bp binding site formed by each double Z-probe; amplifiers then bind to the multiple binding sites on each preamplifier; and finally, labeled probes containing a fluorescent molecule bind to multiple sites of each amplifier. RNAscope data was quantified on sections from coded mice. Each section was viewed and analyzed using CaseViewer 2.4 (3DHISTECH, Budapest, Hungary) or QuPath v.0.2.3 (University of Edinburgh, UK). The *Atlas for the Mouse Brain in Stereotaxic Coordinates* (*Paxinos and Franklin, 2007*) was used to identify every nucleus or subnucleus in which we manually counted *Tshr*, *Lhcgr*, or *Fshr* transcripts in every tenth section using a tag feature. Repeat counting of the same section agreed within <2%. Receptor density was calculated by dividing transcript count by the total area (μm², ImageJ) of each

**Table 1.** Known functions of thyroid-stimulating hormone receptor (TSHR), follicle-stimulating hormone receptor (FSHR), and luteinizing hormone/human chorionic gonadotropin receptor (LHCGR) in brain.

| Receptor | Species | Brain region | Possible function | Reference |
|---|---|---|---|---|
| | Rat | Hypothalamus | Aging | *Emanuele et al., 1985* |
| | Mice | Hippocampus | Spatial learning and memory | *Luan et al., 2020* |
| | Rat | Hypothalamus, hippocampus, pyriform and postcingulate cortex | Thyroid regulation | *Crisanti et al., 2001* |
| | Rat | Hypothalamus | Feeding behavior | *Burgos et al., 2016* |
| | Human | Hypothalamus, amygdala, cingulate gyrus, frontal cortex, hippocampus, thalamus | Mood disorders | *Naicker and Naidoo, 2018* |
| TSHR | Quail | Hypothalamus | Seasonal reproduction | *Williams, 2011* |
| | Yak | Hypothalamus, pineal gland | Follicle growth, maturation, estrus | *Huo et al., 2017* |
| FSHR | Mice | Hippocampus | Mood regulation | *Bi et al., 2020* |
| | Rat | Hypothalamus | Aging | *Emanuele et al., 1985* |
| | Mice | Hippocampus, cortex | Spatial memory, cognition, plasticity | *Blair et al., 2019* |
| | Rat | Hippocampus | Brain metabolism | *Liu et al., 2007* |
| | Fish | Hypothalamus | Functional roles | *Peng et al., 2018* |
| | Mice | Hippocampus | Promote amyloid-β formation | *Lin et al., 2010* |
| | Mice | Cortex | Regulation of neurosteroid production | *Apaja et al., 2004* |
| | Mice | Hypothalamus, hippocampus, midbrain, cortex | Regulation of reproductive functions | *Hämäläinen et al., 1999* |
| | Yak | Hypothalamus, pineal gland | Follicle growth, maturation, estrus | *Huo et al., 2017* |
| LHCGR | Rat | Hypothalamus, hippocampus, dentate gyrus, cerebellum, brainstem, cortex | Cognitive function (Alzheimer's disease) | *Lei et al., 1993* |

region, nucleus or subnucleus. Photomicrographs were prepared using Photoshop CS5.1 (Adobe) only to adjust brightness, contrast, and sharpness, remove artifacts (i.e., obscuring bubbles), and make composite plates.

*Tshr* was detected bilaterally in 173 brain nuclei and subnuclei, in the following descending order of transcript densities: ventricular region, olfactory bulb, forebrain, hypothalamus, medulla, cerebellum, midbrain and pons, cerebral cortex, hippocampus, and thalamus (*Figure 1A*, *Figure 1— source data 1*). Importantly, thyroid glands from *Tshr*$^{-/-}$ mice did not show a signal, proving probe specificity (*Figure 1B*). *Tshr* expression in pooled brain samples was confirmed by qPCR (*Figure 1C*, *Figure 1—source data 2*). The hypothalamus and hippocampus expressed *Tshr*, with hypothalamic expression being considerably higher (p<0.01) in females than in males. Furthermore, within other regions of the brain, highest *Tshr* densities were as follows: ependymal layer of the third ventricle (slightly higher than the thyroid follicular cells); VTT in the olfactory bulb; HDB in the forebrain; MTu in the hypothalamus; SolV in the medulla; PFl in the cerebellum; LDTg in midbrain and pons; DP in the cerebral cortex; DG in hippocampus; and PPT in the thalamus (*Figure 1D*, *Figure 1—source data*

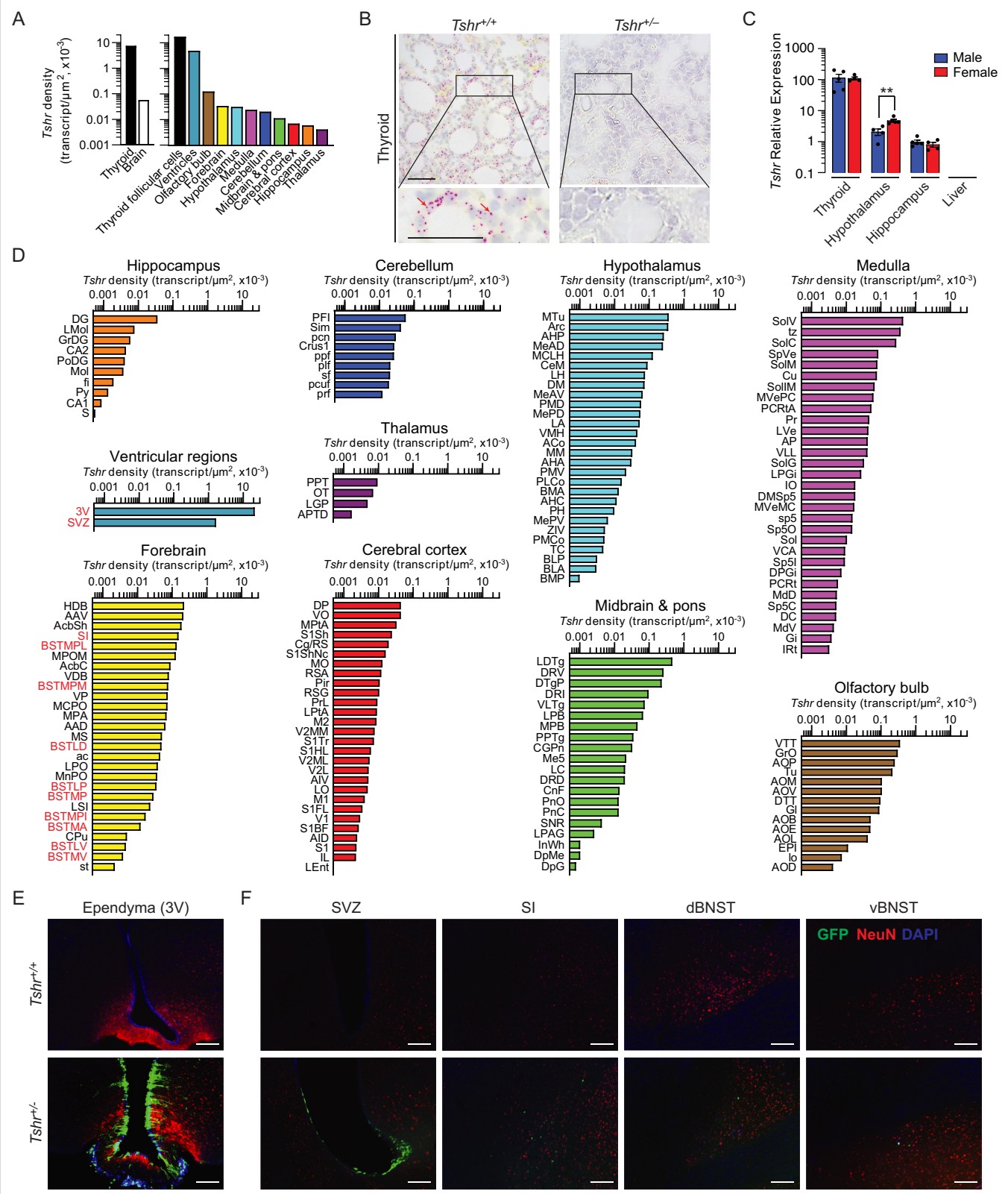

**Figure 1.** *Tshr* expression in the mouse brain. (**A**) *Tshr* transcript density in the thyroid and various brain regions detected by RNAscope. (**B**) RNAscope probe specificity is confirmed in the *Tshr*[+/+] thyroid. *Tshr*[-/-] thyroid was used as negative control. Scale bar: 50 μm. (**C**) *Tshr* expression in the mouse hypothalamus and hippocampus using quantitative PCR. The thyroid and liver serve as positive and negative controls, respectively. Statistics: mean ± SEM, N = 4–5 mice/group, **p<0.01. Data were analyzed by two-tailed Student's *t*-test using Prism v.9.3.1 (GraphPad, San Diego, CA). Significance was

*Figure 1 continued on next page*

*Figure 1 continued*

set at p<0.05. (**D**) *Tshr* transcript density in nuclei and subnuclei of the ventricular regions, olfactory bulb, forebrain, hypothalamus, medulla, cerebellum, midbrain and pons, cerebral cortex, hippocampus, and thalamus. (**E**) Abundant GFP immunofluorescence (green) was detected in the ependymal layer of the third ventricle in *Tshr*[+/−] heterozygous mice, in which a GFP cassette replaced exon 1 of the *Tshr* gene. This GFP signal was absent in *Tshr*[+/+] mice. (**F**) GFP immunofluorescence was also detected in the subventricular zone (SVZ) of the lateral ventricle, and substantia innominata (SI) and dorsal and ventral bed nucleus of stria terminalis (BNST) in the forebrain of the *Tshr*[+/−] mice. Sections were co-stained with DAPI (blue) and a neuronal marker, NeuN (red). Scale bar: 100 μm.

The online version of this article includes the following source data and figure supplement(s) for figure 1:

**Source data 1.** *Tshr* density in brain regions, nuclei, and subnuclei.

**Source data 2.** *Tshr* mRNA expression levels in mouse tissues (qPCR).

**Figure supplement 1.** Raw *Tshr* transcript counts in each brain region, nuclei, and subnuclei.

**Figure supplement 1—source data 1.** *Tshr* transcript count in brain regions, nuclei, and subnuclei.

**Figure supplement 2.** Representative RNAscope micrographs showing *Tshr* transcripts in various regions of the brain.

---

*1*; see Appendix 1 for nomenclature). Raw transcript counts in each region and representative micrographs are shown in *Figure 1—figure supplement 1* (*Figure 1—figure supplement 1—source data 1*) and *Figure 1—figure supplement 2*, respectively.

For purposes of replicability, we employed a complementary approach to study brain *Tshr* expression—the *Tshr*-deficient mouse—in which exon 1 of the *Tshr* gene is replaced by a *Gfp* cassette. This reporter strategy allows for the in vivo display of *Tshr* locations using GFP immunoreactivity (GFP-ir) as a surrogate for *Tshr* expression (*Abe et al., 2003*). Of note is that the *Tshr*[+/−] (haploinsufficient) mouse has one *Tshr* allele intact with normal thyroid function but expresses GFP *in lieu* of one lost allele. In contrast, the *Tshr*[+/+] mouse does not express GFP-ir because both *Tshr* copies are intact and are therefore our negative control.

Consistent with our RNAscope finding, profound GFP-ir was noted in the ependymal region of the third ventricle, mostly in NeuN-negative cells, but with some neuronal localization (*Figure 1E*). The SVZ of the lateral ventricles, and the SI, and dorsal and ventral BNST of the forebrain also showed GFP-ir, but immunoreactivity was much lower than the ependymal layer of the third ventricle (*Figure 1F*). In all, while there was overall concordance between the two methodologies for high *Tshr*-expressing areas, GFP-ir was not detected in a number of *Tshr*-positive regions. This latter discrepancy most likely reflects the grossly lower sensitivity of immunohistochemical detection.

There is evidence that high LH levels in postmenopausal women correlate with a higher incidence of Alzheimer's disease (AD) (*Henderson et al., 1994*; *Rocca et al., 2007*); LHβ transgenic mice are cognitively impaired *Casadesus et al., 2007*; LH receptors (LHCGR) are present in the hippocampus (*Rao, 2017*; *Liu et al., 2007*); and hCG induces cognitive deficits in rodents (*Berry et al., 2008*; *Barron et al., 2010*). Thus, we mapped *Lhcgr* in mouse brain to document expression in 401 brain nuclei and subnuclei. Probe specificity was established by a positive signal in testicular Leydig cells, and with an absent signal in juxtaposed Sertoli cells (*Figure 2A*). Notably similar to *Tshr* transcripts, the ventricular regions displayed the highest transcript density (*Figure 2B*, *Figure 2—source data 1*). Among the brain divisions, the densities were as follows: OV in the ventricular region; SFO in the forebrain; PFI in the cerebellum; MiA in the olfactory bulb; SCO in the thalamus; PMD in the hypothalamus; MVPO in the medulla; DT in midbrain and pons; GrDG in the hippocampus; and SL in the cerebral cortex (*Figure 2C*, *Figure 2—source data 1*). Raw transcript counts in each region and representative micrographs are shown in *Figure 2—figure supplement 1* (*Figure 1—figure supplement 1—source data 1*) and *Figure 2—figure supplement 2*, respectively.

We recently reported the expression of FSHR in mouse, rat, and human brains, particularly in AD-vulnerable regions, including hippocampus and cortex (*Xiong et al., 2022*). We also found that FSH exacerbated AD-like neuropathology and cognitive decline in *3xTg*, *APP/PS1*, and *APP*-KI mice, while the inhibition of FSH action rescued this phenotype. Most notably, shRNA-mediated knockdown of the *Fshr* in the hippocampus prevented the onset of AD-like features (*Xiong et al., 2022*). Here, using RNAscope, we report the expression of *Fshr* at the single-transcript resolution in 353 brain nuclei and subnuclei—and suggest that FSHR in the brain may have roles beyond cognition. Probe specificity was established by a positive signal in testicular Sertoli cells, and an absent signal in

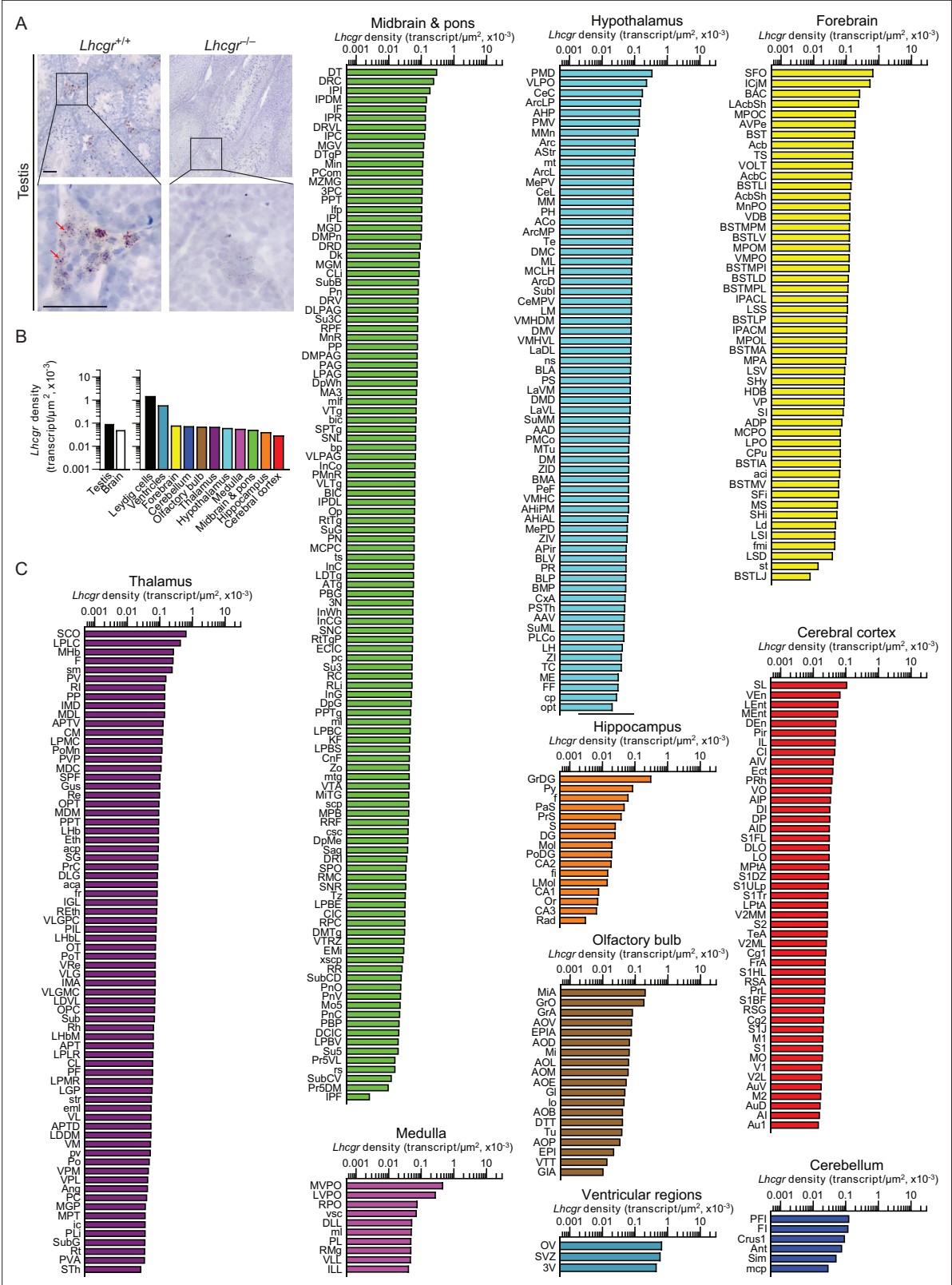

**Figure 2.** *Lhcgr* expression in the mouse brain. (**A**) RNAscope signals were detected in the Leydig cells, but not juxtaposed Sertoli cells, in the mouse testis, confirming probe specificity. Scale bar: 25 μm. (**B**) *Lhcgr* transcript density in the testis and various brain regions detected by RNAscope. (**C**) *Lhcgr* transcript density in nuclei and subnuclei of the ventricular regions, forebrain, cerebellum, olfactory bulb, thalamus, hypothalamus, medulla, midbrain and pons, hippocampus and cerebral cortex.

*Figure 2 continued on next page*

*Figure 2 continued*

The online version of this article includes the following source data and figure supplement(s) for figure 2:

**Source data 1.** *Lhcgr* density in brain regions, nuclei, and subnuclei.

**Figure supplement 1.** Raw *Lhcgr* transcript counts in each brain region, nuclei, and subnuclei.

**Figure supplement 1—source data 1.** *Lhcgr* transcript count in brain regions, nuclei, and subnuclei.

**Figure supplement 2.** Representative RNAscope micrographs showing *Lhcgr* transcripts in various regions of the brain.

juxtaposed Leydig cells and in the testes of *Fshr⁻/⁻* mice—as negative controls (**Figure 3A**). Immuno-fluorescence confirmed the expression of FSHR in NeuN-positive neurons, but not in GFAP-positive glial cells or IBA1-positive microglia (**Figure 3B**).

*Fshr* transcript density was highest in the ventricular region, followed, in descending order, by the cerebellum, olfactory bulb, hippocampus, cerebral cortex, medulla, midbrain and pons, forebrain, thalamus, and hypothalamus (**Figure 3C**, **Figure 3—source data 1**). Within each region, respectively, the highest transcript densities were as follows: ependymal layer of the third ventricle (slightly higher than the testicular Sertoli cells); PFl in the cerebellum; GrA in the olfactory bulb; GrDG in the hippocampus; AIV in the cerebral cortex; RMg in the medulla; MHb in the thalamus; IPDL in midbrain and pons; aci in the forebrain; and ArcL in the hypothalamus (**Figure 3D**, **Figure 3—source data 1**). Raw transcript counts in each region and representative micrographs are shown in **Figure 3—figure supplement 1** (**Figure 3—figure supplement 1—source data 1**) and **Figure 3—figure supplement 2**, respectively.

We used ViewRNA to examine the expression of *FSHR* transcripts in specific regions of the human brain (**Figure 4A**). Expression was noted in neuronal cells co-expressing the noncoding RNA *MALAT1* in the GrDG—consistent with the RNAscope data in mouse brain—and in the parahippocampal cortex. This latter data is consistent with *FSHR* expression in a population of excitatory glutamatergic neurons noted in human brain by 10× single-cell RNA-seq (Allen Brain Atlas). Affymetrix microarray analysis confirmed *FSHR* expression in the frontal, cingulate, temporal, parietal, and occipital subregions of human cortex in postmortem normal and AD brains (**Figure 4B**, **Figure 4—source data 1**). Interestingly, *FSHR* expression trended to be higher in the frontal cortex of the AD brains compared to that of unaffected brains (p=0.060). In all, the data suggest that, beyond a primary role in regulating cognition, brain FSHR may have a wider role in the central regulation.

## Discussion

The past decade has witnessed the unraveling of nontraditional physiological actions of anterior pituitary glycoprotein hormones, and hence, the unmasking of functional receptors in bone, fat, brain, and immune cells, among other organs (**Zaidi et al., 2018**; **Sun et al., 2006**; **Liu et al., 2017**; **Liu et al., 2015**; **Williams, 2011**; **Sun et al., 2020**; **Fields and Shemesh, 2004**). We report here for the first time that *Tshr*, *Lhcgr*, and *Fshr* are expressed in multiple brain regions. The data provide new insights into the distributed central neural network of anterior pituitary hormone receptors, particularly in relation to their role in regulating the somatic tissue function. Specifically, we find a surprising and striking overlap in central neural distribution of the three receptors—with highest transcript densities in the ventricular regions. Furthermore, at least for the TSHR and FSHR, expression levels in ependymal layer of the third ventricle was similar to that of the thyroid follicular cells and testicular Sertoli cells, respectively. *Albeit* intriguing, this may suggest a primary role for these receptors in central neural regulation.

Among 173 *Tshr*-positive brain regions, subregions, and nuclei, the ependymal layer of the third ventricle displayed the highest *Tshr* transcript number and density. This region is juxtaposed to the anterior pituitary that produces TSH in response to hypothalamic TRH. Furthermore, TSH has been reported to be expressed in the hypothalamus (**DeVito et al., 1986**; **Hojvat et al., 1983**). It is therefore possible that a yet uncharacterized central TSH–TSHR feedback circuit may directly regulate the hypothalamic–pituitary–thyroid axis, thought solely to be controlled by thyroid hormones. To add to this complexity, thyroxine-to-triiodothyronine conversion occurs in tanycytes (**Fonseca et al., 2013**), which calls into question whether central TSH actions regulate thyroid hormone metabolism in these

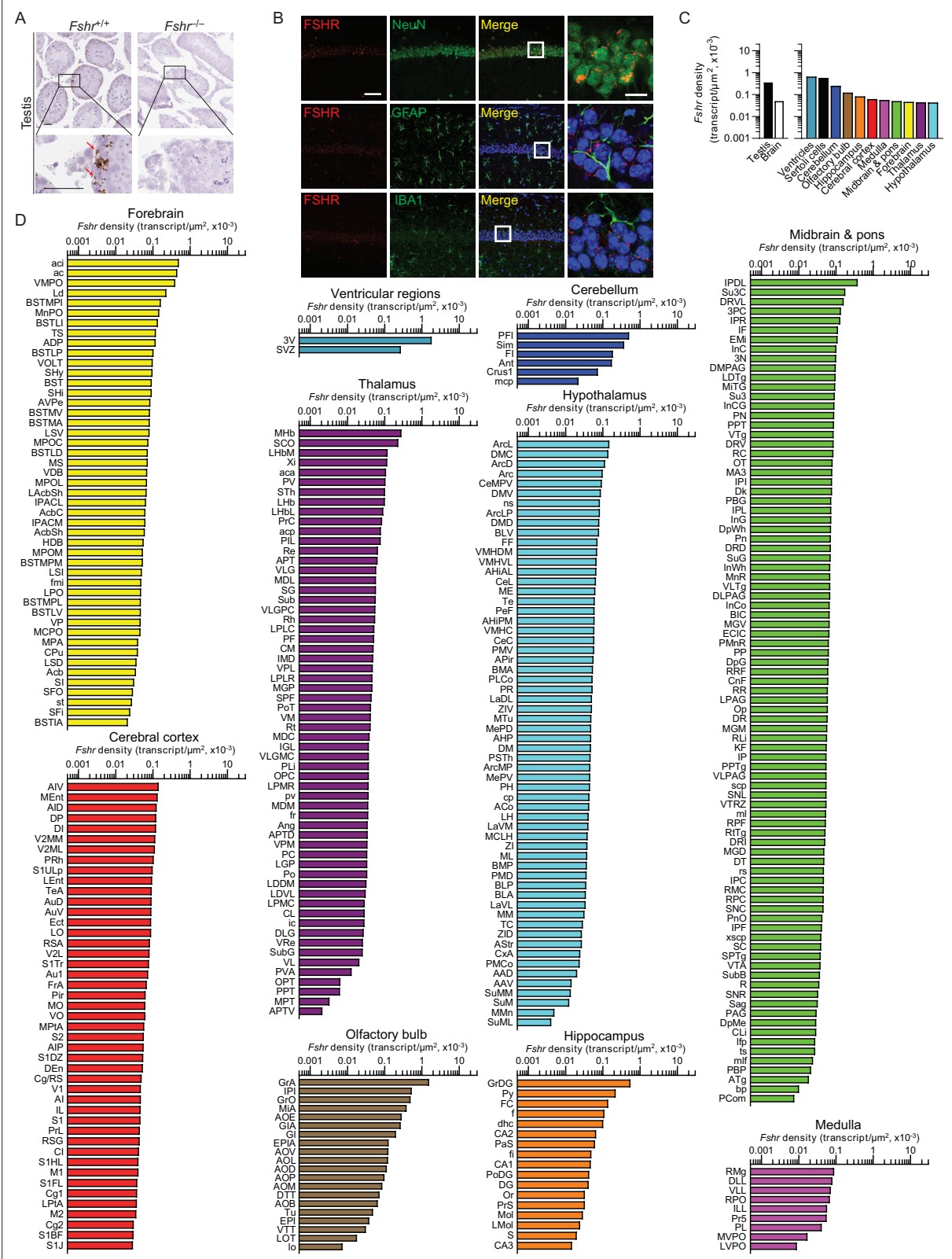

**Figure 3.** *Fshr* expression in the mouse brain. (**A**) RNAscope signals were detected in the Sertoli cells, but not juxtaposed Leydig cells, in the mouse testis, confirming probe specificity. Scale bar: 50 μm. (**B**) Follicle-stimulating hormone receptor (FSHR) immunofluorescence (red) was colocalized with NeuN-positive neurons, but not with GFAP-positive glial cells or IBA1-positive microglia. Scale bar: 100 μm (magnified view, 10 μm). (**C**) *Fshr* transcript

*Figure 3 continued on next page*

*Figure 3 continued*

density in the testis and various brain regions detected by RNAscope. (**D**) *Fshr* transcript density in nuclei and subnuclei of the ventricular regions, cerebellum, olfactory bulb, hippocampus, cerebral cortex, medulla, midbrain and pons, forebrain, thalamus, and hypothalamus.

The online version of this article includes the following source data and figure supplement(s) for figure 3:

**Source data 1.** *Fshr* density in brain regions, nuclei and subnuclei.

**Figure supplement 1.** Raw *Fshr* transcript counts in each brain region, nuclei, and subnuclei.

**Figure supplement 1—source data 1.** *Fshr* transcript count in brain regions, nuclei, and subnuclei.

**Figure supplement 2.** Representative RNAscope micrographs showing *Fshr* transcripts in various regions of the brain.

cells and/or directly modulate hypothalamic TRH neuronal projections. Interestingly, it has been shown that *Tshr* expression is not different between young and old mice (***Kerp et al., 2019***). However, there is conflicting evidence for the expression of TSH with age—with evidence of no difference between 6-, 15-, and 22-month-old mice (***Wang et al., 2019***), but a 44% increase in the old rat compared with the young rat (***Miler et al., 2019***).

The forebrain and olfactory bulb also displayed abundant *Tshr* transcripts, with the highest density in the nucleus of the horizontal limb of the diagonal band (HDB) of the forebrain and ventral tenia tecta (VTT) of the olfactory bulb. These regions are involved, respectively, in learning and odor processing (***Shiotani et al., 2020***; ***Cleland and Linster, 2019***; ***McNamara et al., 2004***; ***Chaves-Coira et al., 2018***; ***Zhan et al., 2013***). In the hypothalamus, the highest density was found in medial tuberal nucleus (MTu), which controls ingestive behaviors and metabolism (***Luo et al., 2018***). Finally, we found

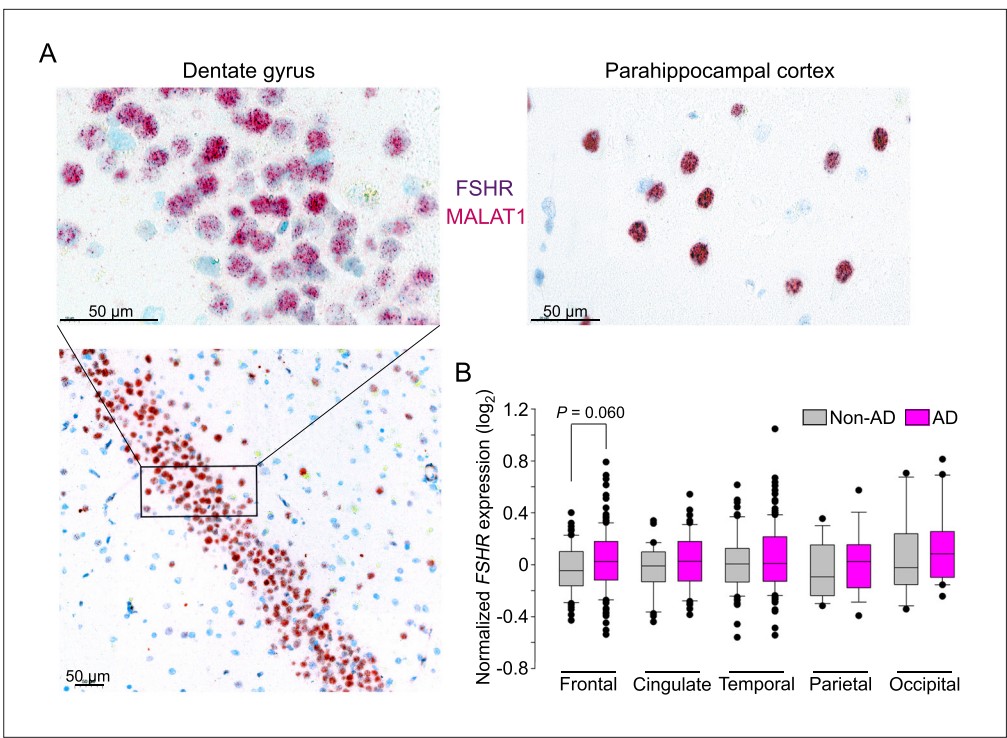

**Figure 4.** *FSHR* expression in the human brain. (**A**) *FSHR* expression in the human hippocampus and parahippocampal cortex was detected by ViewRNA in neuronal cells that coexpress the noncoding RNA *MALAT1*. (**B**) *FSHR* mRNA expression in the frontal, cingulate, temporal, parietal, and occipital subregions of human cortex in postmortem normal and Alzheimer's disease (AD) brains (Affymetrix microarray, from GEO accession: GSE84422). Statistics: mean ± SEM, N = 2–15group, Data were analyzed by two-tailed Student's *t*-test using Prism v.9.3.1 (GraphPad, San Diego, CA).

The online version of this article includes the following source data for figure 4:

**Source data 1.** *FSHR* mRNA expression in the frontal, cingulate, temporal, parietal, and occipital subregions of human cortex in postmortem normal and Alzheimer's disease (AD) brains.

more recently that the modulation of TSHRs in the bed nucleus of the stria terminalis (BNST), which receives direct afferents from the MTu (*Dong and Swanson, 2006*), influences anxiety responses, suggesting that TSHR signaling might, in fact, mediate psychosocial behaviors.

While LH has a key role in reproduction and sexual development, we found 401 brain regions, subregions, and nuclei expressing *Lhcgr*. There were nominal differences in *Lhcgr* expression in many brain regions, but the ventricles stood out as having the highest *Lhcgr* density. Two regions deserve special mention. The *Lhcgr*–rich mitral cell layer of the accessory olfactory bulb (MiA) has a known role in scent communication during mating (*Gildersleeve et al., 2012*; *Lydell and Doty, 1972*; *Huck and Banks, 1984*; *Singh and Bronstad, 2001*). A growing body of evidence suggests that men are attracted to cues of impending ovulation in women, raising an intriguing question on whether cycling hormones affect men's attraction and sexual behavior (*Gildersleeve et al., 2012*; *Singh and Bronstad, 2001*). The broader question is whether LH surges in women during cycling may, in fact, alter male sexual behavior through central mechanisms. Second, a high *Lhcgr* density in the subfornical organ (SFO) of the forebrain was surprising. SFO sends efferent projections to the organum vasculosum of the lamina terminalis (OVLT) (*Miselis, 1981*; *Lind, 1986*), which is surrounded by GnRH neurons and contains estrogen receptors (ESR) (*Low, 2016*). We therefore speculate that circumventricular interactions between LHCGR, LH, GnRH, and ESR underpin the central regulation of reproduction.

RNAscope revealed 353 *Fshr*-expressing brain regions, subregions, and nuclei. Highest expression was noted in the ependymal layer, not surprisingly given its anatomical proximity to the anterior pituitary gland where FSH is produced in response to hypothalamic gonadotropin-releasing hormone (GnRH). The functional significance of *Fshr* expressed in the cerebellum, particularly in the paraflocculus (PFI), is yet unknown. However, other *Fshr*-high subregions, including the granular cell layer of the accessory olfactory bulb (GrA), granular layer of the dentate gyrus (GrDG), and agranular insular cortex (AIV), have known associations with odor processing, learning, memory formation, and anticipation of reward (*Eichenbaum, 2001*; *Nagayama et al., 2014*; *Kesner and Gilbert, 2007*). It is possible that the anosmia of Kallman syndrome, with unclear etiology, may arise from a dysfunctional FSHR-olfaction circuitry. We also find that inactivation of the hippocampal *Fshr* blunts the cognitive impairment and AD-like neuropathology induced by ovariectomy in *3xTg* mice. This data, together with gain- and loss-of-function studies, suggests that hippocampal and cortical FSHR could represent therapeutic targets for AD.

In all, our results provide compelling evidence for multiple central nodes being targets of the anterior pituitary glycoprotein hormones—a paradigm shift that does not conform with the dogma that pituitary hormones are solely master regulators of single bodily processes. Through the intercession of emerging technologies, we compiled the most complete atlas of glycoprotein hormone receptor distribution in the brain at a single-transcript resolution. In addition, we have identified brain sites with the highest transcript expression and density, findings that are imperative toward a better understanding of the neuroanatomical and functional basis of pituitary hormone signaling in the brain. This understanding should provide the foundation for innovative pharmacological interventions for a range of human diseases, wherein direct actions of pituitary hormones have been implicated, importantly, AD.

## Methods

### Mice

We used *Tshr*$^{+/-}$ (strain #004858, Jackson Laboratory), *Lhcgr*$^{-/-}$ (strain #027102, Jackson Laboratory), *Fshr*$^{-/-}$ mice (*Dierich et al., 1998*), and their wild-type littermates in this study. Adult male mice (~3–4-month-old) were housed in a 12 hr:12 hr light:dark cycle at 22 ± 2°C with ad libitum access to water and regular chow. All procedures were approved by the Mount Sinai Institutional Animal Care and Use Committee (approval number IACUC-2018-0047) and are in accordance with Public Health Service and United States Department of Agriculture guidelines.

### RNAscope

Mouse brain tissue was collected for RNAscope. Briefly, mice were anesthetized with isoflurane (2—3% in oxygen; Baxter Healthcare, Deerfield, IL) and transcardially perfused with 0.9% heparinized saline followed by 4% paraformaldehyde (PFA). Brains were extracted and post-fixed in 4% PFA for

24 hr, dehydrated, and embedded into paraffin. Coronal sections were cut at 5 µm, with every tenth section mounted onto ~20 slides with 2–6 sections on each slide. This method allowed to cover the entire brain and eliminate the likelihood of counting the same transcript twice. Sections were air-dried overnight at room temperature and stored at 4°C until required.

Simultaneous detection of mouse *Tshr*, *Lhcgr,* and *Fshr* was performed on paraffin sections using RNAscope 2.5 LS Multiplex Reagent Kit and RNAscope 2.5 LS Probes, namely, Mm-TSHR, Mm-L-HCGR, and Mm-FSHR (Advanced Cell Diagnostics, ACD). RNAscope assays on thyroid glands and testes (positive controls for *Tshr* and *Lhcgr*/*Fshr*, respectively), as well as brains from knockout mice (negative controls), were performed in parallel.

Slides were baked at 60°C for 1 hr, deparaffinized, incubated with hydrogen peroxide for 10 min at room temperature, pretreated with Target Retrieval Reagent for 20 min at 100°C and with Protease III for 30 min at 40°C. Probe hybridization and signal amplification were performed as per the manufacturer's instructions for chromogenic assays.

Following RNAscope assay, the slides were scanned at ×20 magnification and the digital image analysis was successfully validated using the CaseViewer 2.4 (3DHISTECH) software. The same software was employed to capture and prepare images for the figures in the article. Detection of *Tshr*-, *Lhcgr*-, and *Fshr*-positive cells was also performed using the QuPath-0.2.3 (University of Edinburgh, UK) software based on receptor intensity thresholds, size, and shape.

## Histology and immunofluorescence

Heterozygous *Tshr*$^{+/-}$ in which a GFP cassette replaced exon 1 of the *Tshr* gene and their *Tshr*$^{+/+}$ littermates were euthanized with carbon dioxide and perfused transcardially with 0.9% heparinized saline followed by 4% PFA in 0.1 M phosphate-buffered saline (PBS; pH 7.4). Brains were collected and post-fixed in the same fixative overnight at 4°C, then transferred to a 30% sucrose solution in 0.1 M PBS with 0.1% sodium azide and stored at 4°C until they were sectioned on a freezing stage sliding microtome at 30 µm. Sections were stored in 0.1 M PBS solution with 0.1% sodium azide until processed for double immunofluorescence.

For the double-label fluorescent immunohistochemistry, free-floating brain sections were rinsed in 0.1 M PBS (2 × 15 min), followed by a 30 min blocking in 3% normal horse serum (Vector Laboratories, Burlingame, CA) and 0.3% Triton X-100 in 0.1 M PBS. Sections were incubated with a mixture of primary rabbit anti-GFP antibody (1:500; Cat# SP3005P, OriGene, Rockville, MD) and mouse anti-NeuN antibody (1:1000; Cat# ab104224, Abcam, Cambridge, MA) for 18 hr. Sections were then incubated with the secondary donkey anti-rabbit Alexa 488 (1:700; Cat# 711-545-152, Jackson ImmunoResearch, West Grove, PA) and donkey anti-mouse DyLight 594 (1:700; Cat# DK-2594, Vector Laboratories) antibodies in 0.1 M PBS for 3 hr at room temperature. For immunohistochemical controls, the primary antibody was either omitted or pre-adsorbed with the immunizing peptide overnight at 4°C, resulting in no immunoreactive staining. In addition, we expectedly did not detect GFP immunoreactivity (-ir) in the *Tshr*$^{+/+}$ littermates as the *Tshr* gene was intact and did not express GFP. Sections were mounted onto slides (Superfrost Plus) and cover-slipped using ProLong Gold Antifade Reagent (Life Technologies, Grand Island, NY). All steps were performed at room temperature.

For immunofluorescence staining for FSHR, free-floating brain sections were incubated overnight at 4°C with primary anti-FSHR (1:200; Cat# PA5-50963, Thermo Fisher), anti-NeuN (1:300; Cat# MAB377, Sigma-Aldrich), anti-GFAP (1:400; Cat# MAB360, Sigma-Aldrich), or anti-IBA1 (1:500; Cat# PA5-18039, Thermo Fisher) antibodies. After washing with Tris-buffered saline, the sections were incubated with a mixture of labeled secondary antibodies for detection. DAPI (Sigma-Aldrich) was used for staining nuclei.

## Microarray analysis

Affymetrix Human Genome U133 Plus 2.0 Array data for *FSHR* expression in the frontal, cingulate, temporal, parietal, and occipital cortex from both AD and non-AD human brains were curated from a previously published dataset (GEO accession #GSE84422; *Wang et al., 2016*).

## Quantitative PCR

For quantitative RT-PCR performed on homogenates of brain tissues, total RNA from the hypothalamus and the hippocampus isolated from five *Tshr*$^{+/+}$ mice was extracted using an RNeasy Mini kit

(QIAGEN) as per the manufacturer's protocol. Thyroid and liver tissues were used as positive and negative controls, respectively. RNA was treated with DNAse I (Invitrogen), and reverse-transcribed using the SuperScript II Reverse Transcriptase (Thermo Fisher Scientific). qPCR was performed with a QuantStudio 7 Real-Time PCR system (Applied Biosystems). PCR reaction mix consisted of first-strand cDNA template, exon-spanning primer pairs, and SYBR Green PCR master mix (Thermo Fisher Scientific). Expression of the selected targets was compared to that of a panel of normalizing genes (*Rps11*, *Tubg1,* and *Gapdh*) measured on the same sample in parallel on the same plate, giving a Ct difference ($\Delta$Ct) for the normalizing gene minus the test gene. Relative expression levels were calculated by $2^{-\Delta\Delta Ct}$ using thyroid as the reference tissue.

## Quantitation, validation, and statistical analysis

Immunofluorescent images were viewed and captured using ×10 or ×20 objectives with an Observer. Z1 fluorescence microscope (Carl Zeiss, Germany) with appropriate filters for Alexa 488, Cy3, and DAPI. The captured GFP and NeuN images were evaluated and overlaid using AxioVision v.4.8 software (Carl Zeiss, Germany) and ImageJ (NIH, Bethesda, MD).

Data were analyzed by two-tailed Student's *t*-test using Prism v.9.3.1 (GraphPad, San Diego, CA). Significance was set at $p < 0.05$.

## Acknowledgements

Work at the Icahn School of Medicine at Mount Sinai was performed at the Center for Translational Medicine and Pharmacology and was supported by U19 AG060917 to MZ and CJR; R01 DK113627 to MZ and TFD; R01 AG074092 and U01 AG073148 to TY and MZ; and R01 AG071870 to S-MK, TY and MZ. MZ also thanks the Harrington Discovery Institute for the Innovator-Scholar Award. CJR acknowledges support from the NIH (P20 GM121301).

## Additional information

### Competing interests

Vahram Haroutunian: has received consultation fees from Synaptec to Cold Spring Harbor Laboratories. Keqiang Ye, Tony Yuen: Reviewing editor, *eLife*. Terry F Davies: has received payments from Kronus Inc, Starr, ID as a Board member and for various books and ebooks. Mone Zaidi: Senior editor, *eLife*. The other authors declare that no competing interests exist.

### Funding

| Funder | Grant reference number | Author |
| --- | --- | --- |
| National Institute on Aging | U19 AG060917 | Clifford J Rosen<br>Mone Zaidi |
| National Institute of Diabetes and Digestive and Kidney Diseases | R01 DK113627 | Terry F Davies |
| National Institute on Aging | R01 AG074092 | Tony Yuen<br>Mone Zaidi |
| National Institute on Aging | U01 AG073148 | Tony Yuen<br>Mone Zaidi |
| National Institute on Aging | R01 AG071870 | Se-Min Kim<br>Tony Yuen<br>Mone Zaidi |
| National Institute of General Medical Sciences | P20 GM121301 | Clifford J Rosen |

The funders had no role in study design, data collection and interpretation, or the decision to submit the work for publication.

## Author contributions
Vitaly Ryu, Data curation, Formal analysis, Validation, Investigation, Writing - original draft; Anisa Gumerova, Pavel Katsel, Liam Cullen, TanChun Kuo, Data curation, Investigation; Funda Korkmaz, Sari Miyashita, Hasni Kannangara, Ashley Padilla, Farhath Sultana, Soleil A Wizman, Natan Kramskiy, Se-Min Kim, Ki A Goosens, Investigation; Seong Su Kang, Pokman Chan, Data curation, Investigation, Methodology; Samir Zaidi, Data curation; Maria I New, Vahram Haroutunian, Keqiang Ye, Conceptualization, Project administration; Clifford J Rosen, Terry F Davies, Conceptualization, Funding acquisition, Project administration; Tal Frolinger, Validation, Investigation; Daria Lizneva, Investigation, Project administration; Tony Yuen, Conceptualization, Supervision, Funding acquisition, Methodology, Writing - original draft, Project administration, Writing - review and editing; Mone Zaidi, Conceptualization, Supervision, Funding acquisition, Writing - original draft, Project administration, Writing - review and editing

## Author ORCIDs
Vitaly Ryu (iD) http://orcid.org/0000-0001-8068-4577
Seong Su Kang (iD) http://orcid.org/0000-0002-2517-9962
Pavel Katsel (iD) http://orcid.org/0000-0001-8076-0162
TanChun Kuo (iD) http://orcid.org/0000-0001-5301-755X
Ki A Goosens (iD) http://orcid.org/0000-0002-5246-2261
Mone Zaidi (iD) http://orcid.org/0000-0001-5911-9522

## Ethics
All procedures were approved by the Mount Sinai Institutional Animal Care and Use Committee (approval number IACUC-2018-0047) and are in accordance with Public Health Service and United States Department of Agriculture guidelines.

## Decision letter and Author response
Decision letter https://doi.org/10.7554/eLife.79612.sa1
Author response https://doi.org/10.7554/eLife.79612.sa2

# Additional files

## Supplementary files
• MDAR checklist

## Data availability
All data generated or analyzed during this study are included in the manuscript and supporting file.

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

# Appendix 1

Glossary of the brain regions, nuclei, and subnuclei.

**Cerebellum**
Ant anterior lobe cerebellum
Crus1 crus 1 of the ansiform lobule
FI flocculus
mcp middle cerebellar peduncle
pcn precentral fissure
pcuf preculminate fissure
PFI paraflocculus
plf posterolateral fissure
ppf prepyramidal fissure
prf primary fissure
sf secondary fissure
Sim simple lobule
**Cerebral cortex**
AI agranular insular cortex
AID agranular insular cortex, dorsal part
AIP agranular insular cortex, posterior part
AIV agranular insular cortex, ventral part
Au1 primary auditory cortex
AuD secondary auditory cortex, dorsal area
AuV secondary auditory cortex, ventral area
Cg/RS cingular/retrosplenial cortex
Cg1 cingulate cortex, area 1
Cg2 cingulate cortex, area 2
CI caudal interstitial nucleus of the medial longitudinal fasciculus
DEn dorsal endopiriform nucleus
DI dysgranular insular cortex
DLO dorsolateral orbital cortex
DP dorsal peduncular cortex
Ect ectorhinal cortex
FrA frontal association cortex
IL infralimbic cortex
LEnt lateral entorhinal cortex
LO lateral orbital cortex
LPtA lateral parietal association cortex
M1 primary motor cortex
M2 secondary motor cortex
MEnt medial entorhinal cortex
MO medial orbital cortex
MPtA medial parietal association cortex
Pir piriform cortex
PRh perirhinal cortex
PrL prelimbic cortex
RSA retrosplenial agranular cortex
RSG retrosplenial granular cortex
S1 primary somatosensory cortex
S1BF primary somatosensory cortex, barrel field
S1DZ primary somatosensory cortex, dysgranular region
S1FL primary somatosensory cortex, forelimb region
S1HL primary somatosensory cortex, hindlimb region
S1J primary somatosensory cortex, jaw region
S1Sh primary somatosensory cortex, shoulder region
S1ShNc primary somatosensory cortex, shoulder/neck region

S1Tr primary somatosensory cortex, trunk region
S1ULp primary somatosensory cortex, upper lip region
S2 secondary somatosensory cortex
SL semilunar nucleus
TeA temporal association cortex
V1 primary visual cortex
V2L secondary visual cortex, lateral area
V2ML secondary visual cortex, mediolateral area
V2MM secondary visual cortex, mediomedial area
VEn ventral endopiriform nucleus
VO ventral orbital cortex

**Forebrain**

AAD anterior amygdaloid area, dorsal part
AAV anterior amygdaloid area, ventral part
ac anterior commissure
Acb accumbens nucleus
AcbC accumbens nucleus, core
AcbSh accumbens nucleus, shell
aci anterior commissure, intrabulbar part
ADP anterodorsal preoptic nucleus
AVPe anteroventral periventricular nucleus
BAC bed nucleus of the anterior commissure
BST bed nucleus of the stria terminalis
BSTIA bed nucleus of the stria terminalis, intraamygdaloid division
BSTLD bed nucleus of the stria terminalis, lateral division, dorsal part
BSTLI bed nucleus of the stria terminalis, lateral division, intermediate part
BSTLJ bed nucleus of the stria terminalis, lateral division, juxtacapsular part
BSTLP bed nucleus of the stria terminalis, lateral division, posterior part
BSTLV bed nucleus of the stria terminalis, lateral division, ventral part
BSTMA bed nucleus of the stria terminalis, medial division, anterior part
BSTMP bed nucleus of the stria terminalis, medial division, posterior part
BSTMPI bed nucleus of the stria terminalis, medial division, posterointermediate part
BSTMPL bed nucleus of the stria terminalis, medial division, posterolateral part
BSTMPM bed nucleus of the stria terminalis, medial division, posteromedial part
BSTMV bed nucleus of the stria terminalis, medial division, ventral part
CPu caudate putamen (striatum)
fmi forceps minor of the corpus callosum
HDB nucleus of the horizontal limb of the diagonal band
ICjM islands of Calleja, major island
IPACL interstitial nucleus of the posterior limb of the anterior commissure, lateral part
IPACM interstitial nucleus of the posterior limb of the anterior commissure, medial part
LAcbSh lateral accumbens shell
Ld lambdoid septal zone
LPO lateral preoptic area
LSD lateral septal nucleus, dorsal part
LSI lateral septal nucleus, intermediate part
LSS lateral stripe of the striatum
LSV lateral septal nucleus, ventral part
MCPO magnocellular preoptic nucleus
MnPO median preoptic nucleus
MPA medial preoptic area
MPOC medial preoptic nucleus, central part
MPOL medial preoptic nucleus, lateral part
MPOM medial preoptic nucleus, medial part
MS medial septal nucleus

SFi septofimbrial nucleus
SFO subfornical organ
SHi septohippocampal nucleus
SHy septohypothalamic nucleus
SI substantia innominata
st stria terminalis
TS triangular septal nucleus
VDB nucleus of the vertical limb of the diagonal band
VMPO ventromedial preoptic nucleus
VOLT vascular organ of the lamina terminalis
VP ventral pallidum

**Hippocampus**
CA1 field CA1 of hippocampus
CA2 field CA2 of hippocampus
CA3 field CA3 of hippocampus
DG dentate gyrus
dhc dorsal hippocampal commissure
f fornix
FC fasciola cinereum
fi fimbria of the hippocampus
GrDG granular layer of the dentate gyrus
LMol lacunosum moleculare layer of the hippocampus
Mol molecular layer of the dentate gyrus
Or oriens layer of the hippocampus
PaS parasubiculum
PoDG polymorph layer of the dentate gyrus
PrS presubiculum
Py pyramidal tract
Rad stratum radiatum of the hippocampus
S subiculum

**Hypothalamus**
AAD anterior amygdaloid area, dorsal part
AAV anterior amygdaloid area, ventral part
ACo anterior cortical amygdaloid nucleus
AHA anterior hypothalamic area, anterior part
AHC anterior hypothalamic area, central part
AHiAL amygdalohippocampal area, anterolateral part
AHiPM amygdalohippocampal area, posteromedial part
AHP anterior hypothalamic area, posterior part
APir amygdalopiriform transition area
Arc arcuate hypothalamic nucleus
ArcD arcuate hypothalamic nucleus, dorsal part
ArcL arcuate hypothalamic nucleus, lateral part
ArcLP arcuate hypothalamic nucleus, lateroposterior part
ArcMP arcuate hypothalamic nucleus, medial posterior part
AStr amygdalostriatal transition area
BLA basolateral amygdaloid nucleus, anterior part
BLP basolateral amygdaloid nucleus, posterior part
BLV basolateral amygdaloid nucleus, ventral part
BMA basolateral amygdaloid nucleus, anterior part
BMP basomedial amygdaloid nucleus, posterior part
CeC central amygdaloid nucleus, capsular part
CeL central amygdaloid nucleus, lateral division
CeM central amygdaloid nucleus, medial division
CeMPV central amygdaloid nucleus, medial posteroventral part

cp cerebral peduncle, basal part
CxA cortex-amygdala transition zone
DM dorsomedial hypothalamic nucleus
DMC dorsomedial hypothalamic nucleus, compact part
DMD dorsomedial hypothalamic nucleus, dorsal part
DMV dorsomedial hypothalamic nucleus, ventral part
FF fields of Forel
LA lateroanterior hypothalamic nucleus
LaDL lateral amygdaloid nucleus, dorsolateral part
LaVL lateral amygdaloid nucleus, ventrolateral part
LaVM lateral amygdaloid nucleus, ventromedial part
LH lateral hypothalamic area
LM lateral mammillary nucleus
MCLH magnocellular nucleus of the lateral hypothalamus
ME median eminence
MeAD medial amygdaloid nucleus, anteriodorsal part
MeAV medial amygdaloid nucleus, anteroventral part
MePD medial amygdaloid nucleus, posterodorsal part
MePV medial amygdaloid nucleus, posteroventral part
ML medial mammillary nucleus, lateral part
MM medial mammillary nucleus, medial part
MMn medial mammillary nucleus, median part
mt mammillothalamic tract
MTu medial tuberal nucleus
ns nigrostriatal bundle
opt optic tract
PeF perifornical nucleus
PH posterior hypothalamic area
PLCo posterolateral cortical amygdaloid nucleus
PMCo posteromedial cortical amygdaloid nucleus
PMD premammillary nucleus, dorsal part
PMV premammillary nucleus, ventral part
PR prerubral field
PS parastrial nucleus
PSTh parasubthalamic nucleus
Subl subincertal nucleus
SuM supramammillary nucleus
SuML supramammillary nucleus, lateral part
SuMM supramammillary nucleus, medial part
TC tuber cinereum area
Te terete hypothalamic nucleus
VLPO ventrolateral preoptic nucleus
VMH ventromedial hypothalamic nucleus
VMHC ventromedial hypothalamic nucleus, central part
VMHDM ventromedial hypothalamic nucleus, dorsomedial part
VMHVL ventromedial hypothalamic nucleus, ventrolateral part
ZI zona incerta
ZID zona incerta, dorsal part
ZIV zona incerta, ventral part
**Medulla**
AP area postrema
Cu cuneate nucleus
DC dorsal cochlear nucleus
DLL dorsal nucleus of the lateral lemniscus
DMSp5 dorsomedial spinal trigeminal nucleus

DPGi dorsal paragigantocellular nucleus
Gi gigantocellular reticular nucleus
ILL intermediate nucleus of the lateral lemniscus
IO inferior olive
IRt intermediate reticular nucleus
LPGi lateral paragigantocellular nucleus
LVe lateral vestibular nucleus
LVPO lateroventral periolivary nucleus
MdD medullary reticular nucleus, dorsal part
MdV medullary reticular nucleus, ventral part
ml medial lemniscus
MVeMC medial vestibular nucleus, magnocellular part
MVePC medial vestibular nucleus, parvicellular part
MVPO medioventral periolivary nucleus
PCRt parvicellular reticular nucleus
PCRtA parvicellular reticular nucleus, alpha part
PL paralemniscal nucleus
Pr prepositus nucleus
Pr5 principal sensory trigeminal nucleus
RMg raphe magnus nucleus
RPO rostral periolivary region
Sol solitary tract
SolC nucleus of the solitary tract, commissural part
SolG nucleus of the solitary tract, gelatinous part
SolIM nucleus of the solitary tract, intermediate part
SolM nucleus of the solitary tract, medial part
SolV solitary nucleus, ventral part
sp5 spinal trigeminal tract
Sp5C spinal trigeminal nucleus, caudal part
Sp5I spinal trigeminal nucleus, interpolar part
Sp5O spinal trigeminal nucleus, oral part
SpVe spinal vestibular nucleus
tz trapezoid body
VCA ventral cochlear nucleus, anterior part
VLL ventral nucleus of the lateral lemniscus
vsc ventral spinocerebellar tract

**Midbrain and pons**
3N oculomotor nucleus
3PC oculomotor nucleus, parvicellular part
ATg anterior tegmental nucleus
BIC nucleus of the brachium of the inferior colliculus
bic brachium of the inferior colliculus
bp brachium pontis (stem of middle cerebellar peduncle)
CGPn central gray of the pons
CIC central nucleus of the inferior colliculus
CLi caudal linear nucleus of the raphe
CnF cuneiform nucleus
csc commissure of the superior colliculus
DCIC dorsal cortex of the inferior colliculus
Dk nucleus of Darkschewitsch
DLPAG dorsolateral periaqueductal gray
DMPAG dorsomedial periaqueductal gray
DMPn dorsomedial pontine nucleus
DMTg dorsomedial tegmental area
DpG deep gray layer of the superior colliculus

DpMe deep mesencephalic nucleus
DpWh deep white layer of the superior colliculus
DR dorsal raphe nucleus
DRC dorsal raphe nucleus, caudal part
DRD dorsal raphe nucleus, dorsal part
DRI dorsal raphe nucleus, interfascicular part
DRV dorsal raphe nucleus, ventral part
DRVL dorsal raphe nucleus, ventrolateral part
DT dorsal terminal nucleus of the accessory optic tract
DTgP dorsal tegmental nucleus, pericentral part
ECIC external cortex of the inferior colliculus
EMi epimicrocellular nucleus
IF interfascicular nucleus
InC interstitial nucleus of Cajal
InCG interstitial nucleus of Cajal, greater part
InCo intercollicular nucleus
InG intermediate gray layer of the superior colliculus
InWh intermediate white layer of the superior colliculus
IP interpeduncular nucleus
IPC interpeduncular nucleus, caudal subnucleus
IPDL interpeduncular nucleus, dorsolateral subnucleus
IPDM interpeduncular nucleus, dorsomedial subnucleus
IPF interpeduncular fossa
IPI interpeduncular nucleus, intermediate subnucleus
IPL interpeduncular nucleus, lateral subnucleus
IPR interpeduncular nucleus, rostral subnucleus
KF Kölliker-Fuse nucleus
LC locus coeruleus
LDTg laterodorsal tegmental nucleus
lfp longitudinal fasciculus of the pons
LPAG lateral periaqueductal gray
LPB lateral parabrachial nucleus
LPBC lateral parabrachial nucleus, central part
LPBE lateral parabrachial nucleus, external part
LPBS lateral parabrachial nucleus, superior part
LPBV lateral parabrachial nucleus, ventral part
MA3 medial accessory oculomotor nucleus
MCPC magnocellular nucleus of the posterior commissure
Me5 mesencephalic trigeminal nucleus
MGD medial geniculate nucleus, dorsal part
MGM medial geniculate nucleus, medial part
MGV medial geniculate nucleus, ventral part
Min minimus nucleus
MiTG microcellular tegmental nucleus
ml medial lemniscus
mlf medial longitudinal fasciculus
MnR median raphe nucleus
Mo5 motor trigeminal nucleus
MPB medial parabrachial nucleus
mtg mammillotegmental tract
MZMG marginal zone of the medial geniculate
Op optic nerve layer of the superior colliculus
OT nucleus of the optic tract
PAG periaqueductal gray
PBG parabigeminal nucleus

PBP parabrachial pigmented nucleus
pc posterior commissure
PCom nucleus of the posterior commissure
PMnR paramedian raphe nucleus
Pn pontine nuclei
PN paranigral nucleus
PnC pontine reticular nucleus, caudal part
PnO pontine reticular nucleus, oral part
PnV pontine reticular nucleus, ventral part
PP peripeduncular nucleus
PPT posterior pretectal nucleus
PPTg pedunculopontine tegmental nucleus
Pr5DM principal sensory trigeminal nucleus, dorsomedial part
Pr5VL principal sensory trigeminal nucleus, ventrolateral part
R red nucleus
RC raphe cap
RLi rostral linear nucleus of the raphe
RMC red nucleus, magnocellular part
RPC red nucleus, parvicellular part
RPF retroparafascicular nucleus
RR retrorubral nucleus
RRF retrorubral field
rs rubrospinal tract
RtTg reticulotegmental nucleus of the pons
RtTgP reticulotegmental nucleus of the pons, pericentral part
Sag sagulum nucleus
SC superior colliculus
scp superior cerebellar peduncle (brachium conjunctivum)
SNC substantia nigra, compact part
SNL substantia nigra, lateral part
SNR substantia nigra, reticular part
SPO superior paraolivary nucleus
SPTg subpedencular tegmental nucleus
Su3 supraoculomotor periaqueductal gray
Su3C supraoculomotor cap
Su5 supratrigeminal nucleus
SubB subbrachial nucleus
SubCD subcoeruleus nucleus, dorsal part
SubCV subcoeruleus nucleus, ventral part
SuG superficial gray layer of the superior colliculus
ts tectospinal tract
Tz nucleus of the trapezoid body
VLPAG ventrolateral periaqueductal gray
VLTg ventrolateral tegmental area
VTA ventral tegmental area
VTg ventral tegmental nucleus
VTRZ visual tegmental relay zone
xscp decussation of the superior cerebellar peduncle
Zo zonal layer of the superior colliculus
**Olfactory bulb**
AOB accessory olfactory bulb
AOD anterior olfactory nucleus, dorsal part
AOE anterior olfactory nucleus, external part
AOL anterior olfactory nucleus, lateral part
AOM anterior olfactory nucleus, medial part

AOP anterior olfactory nucleus, posterior part
AOV anterior olfactory nucleus, ventral part
DTT dorsal tenia tecta
EPl external plexiform layer of the olfactory bulb
EPIA external plexiform layer of the accessory olfactory bulb
GIA glomerular layer of the accessory olfactory bulb
Gl glomerular layer of the olfactory bulb
GrA granule cell layer of the accessory olfactory bulb
GrO granular cell layer of the olfactory bulb
IPI interpeduncular nucleus, intermediate subnucleus
lo lateral olfactory tract
LOT nucleus of the lateral olfactory tract
Mi mitral cell layer of the olfactory bulb
MiA mitral cell layer of the accessory olfactory bulb
Tu olfactory tubercle
VTT ventral tenia tecta

**Thalamus**
aca anterior commissure, anterior part
acp anterior commissure, posterior
Ang angular thalamic nucleus
APT anterior pretectal nucleus
APTD anterior pretectal nucleus, dorsal part
APTV anterior pretectal nucleus, ventral part
CL centrolateral thalamic nucleus
CM central medial thalamic nucleus
DLG dorsal lateral geniculate nucleus
eml external medullary lamina
Eth ethmoid thalamic nucleus
F nucleus of the fields of Forel
fr fasciculus retroflexus
Gus gustatory thalamic nucleus
ic internal capsule
IGL intergeniculate leaf
IMA intramedullary thalamic area
IMD intermediodorsal thalamic nucleus
LDDM laterodorsal thalamic nucleus, dorsomedial part
LDVL laterodorsal thalamic nucleus, ventrolateral part
LGP lateral globus pallidus
LHb lateral habenular nucleus
LHbL lateral habenular nucleus, lateral part
LHbM lateral habenular nucleus, medial part
LPLC lateral posterior thalamic nucleus, laterocaudal part
LPLR lateral posterior thalamic nucleus, laterorostral part
LPMC lateral posterior thalamic nucleus, mediocaudal part
LPMR lateral posterior thalamic nucleus, mediorostral part
MDC mediodorsal thalamic nucleus, central part
MDL mediodorsal thalamic nucleus, lateral part
MDM mediodorsal thalamic nucleus, medial part
MGP medial globus pallidus (entopeduncular nucleus)
MHb medial habenular nucleus
MPT medial pretectal nucleus
OPC oval paracentral thalamic nucleus
OPT olivary pretectal nucleus
OT nucleus of the optic tract
PC paracentral thalamic nucleus

PF parafascicular thalamic nucleus
PIL posterior intralaminar thalamic nucleus
PLi posterior limitans thalamic nucleus
Po posterior thalamic nuclear group
PoMn posteromedian thalamic nucleus
PoT posterior thalamic nuclear group, triangular part
PP peripeduncular nucleus
PPT posterior pretectal nucleus
PrC precommissural nucleus
pv periventricular fiber system
PV paraventricular thalamic nucleus
PVA paraventricular thalamic nucleus, anterior part
PVP paraventricular thalamic nucleus, posterior part
Re reuniens thalamic nucleus
REth retroethmoid nucleus
Rh rhomboid thalamic nucleus
RI rostral interstitial nucleus of medial longitudinal fasciculus
Rt reticular thalamic nucleus
SCO subcommissural organ
SG suprageniculate thalamic nucleus
sm stria medullaris of the thalamus
SPF subparafascicular thalamic nucleus
STh subthalamic nucleus
str superior thalamic radiation
Sub submedius thalamic nucleus
SubG subgeniculate nucleus
VL ventrolateral thalamic nucleus
VLG ventral lateral geniculate nucleus
VLGMC ventral lateral geniculate nucleus, magnocellular part
VLGPC ventral lateral geniculate nucleus, parvicellular part
VM ventromedial thalamic nucleus
VPL ventral posterolateral thalamic nucleus
VPM ventral posteromedial thalamic nucleus
VRe ventral reuniens thalamic nucleus
Xi xiphoid thalamic nucleus
**Ventricular zones**
3V 3rd ventricle
OV olfactory ventricle (olfactory part of lateral ventricle)
SVZ subventricular zone of the lateral ventricle

