## [Editor Report]

This article is an excellent resource as an atlas of hypophyseal hormone localization in the brain. It is an invaluable resource to researchers in the field and provides important new information.

---

## [Decision Letter]

**Decision letter after peer review:**

Thank you for submitting your article "Brain Atlas for Glycoprotein Hormone Receptors at Single-Transcript Level" for consideration by *eLife*. Your article has been reviewed by 3 peer reviewers, and the evaluation has been overseen by a Reviewing Editor and Carlos Isales as the Senior Editor. The following individuals involved in the review of your submission have agreed to reveal their identity: Yunlei Yang (Reviewer #1); Christopher L-H Huang (Reviewer #2); Mei Wan (Reviewer #3).

Essential revisions:

1. As the authors did not stain tanycytes using tanycyte-specific antibodies, I would suggest co-labelling tanycyte and TSHR in Figure 1, or removing the word "tanycyte" in the text but with an alternative statement "ependymal layer of the third ventricle".

2. In this study, the authors used young adult animals. Are there any differences in those gene and receptor expressions in young or old mouse brains? It would gain more information to discuss.

3. Figure 1C- based on the figure legend, n=4-5 mice were used for each group. The bar diagram should be changed into a scatter dot plot format.

4. Figure 4A- the scale bars of the 2 images in the upper panel are not clear.

5. Supplemental Figure 2- the resolution of some higher power images needs to be improved.

---

## [Author Response]

Essential revisions:1. As the authors did not stain tanycytes using tanycyte-specific antibodies, I would suggest co-labelling tanycyte and TSHR in Figure 1, or removing the word "tanycyte" in the text but with an alternative statement "ependymal layer of the third ventricle".

As we do not currently have *Tshr^+/-^* mice and it would take a while to grow these colonies, we have chosen to replace the word “tanycyte” with an “ependymal layer of the third ventricle or cells” in the text.

2. In this study, the authors used young adult animals. Are there any differences in those gene and receptor expressions in young or old mouse brains? It would gain more information to discuss.

We fully concur with the review comment. As for the receptor, Kerp *et al.,* showed no difference in pituitary *Tshr* expression in young *vs*. old mice. For the ligand, while Wang *et al.,* did not detect alterations in TSH levels in 6, 15 and 22–month–old mice, Miler *et al.,* reported an increase in pituitary TSH by 44% in old Wistar rats compared with young rats. These references have been quoted and discussed in lines 226 to 229 in the revised manuscript.

3. Figure 1C- based on the figure legend, n=4-5 mice were used for each group. The bar diagram should be changed into a scatter dot plot format.

Individual data points have been added to the bar graph in Figure 1C.

4. Figure 4A- the scale bars of the 2 images in the upper panel are not clear.

The scale bar has been updated to provide more clarity.

5. Supplemental Figure 2- the resolution of some higher power images needs to be improved.

We have replaced the high–power images as suggested. Notably, in the ventral tenia tecta (VTT) of the olfactory bulb the pattern of TSHR expression differed from other brain sites. The expression appeared densely packed within the neurons of the VTT as compared to the dispersed dotty expression in other regions of the brain.